# Human Heart Explant-Derived Extracellular Vesicles: Characterization and Effects on the In Vitro Recellularization of Decellularized Heart Valves

**DOI:** 10.3390/ijms20061279

**Published:** 2019-03-14

**Authors:** Amanda Leitolis, Paula Hansen Suss, João Gabriel Roderjan, Addeli Bez Batti Angulski, Francisco Diniz Affonso da Costa, Marco Augusto Stimamiglio, Alejandro Correa

**Affiliations:** 1Laboratory of Basic Biology of Stem Cells, Carlos Chagas Institute, Fiocruz-Paraná, Curitiba 81350-010, Brazil; aleitolis@gmail.com (A.L.); addeli.angulski@gmail.com (A.B.B.A.); marco.stimamiglio@fiocruz.br (M.A.S.); 2Pontifical Catholic University of Paraná—PUCPR, Curitiba 80215-901, Brazil; paula.h@pucpr.br (P.H.S.); costa.f@pucpr.br (F.D.A.d.C.); 3Technological Federal University of Paraná—UTFPR, Curitiba 80230-901, Brazil; gabrielrdm@gmail.com

**Keywords:** human heart, tissue explant, extracellular vesicle, heart valve, tissue engineering, recellularization, cardiac regions, mesenchymal stromal cells

## Abstract

Extracellular vesicles (EVs) are particles released from different cell types and represent key components of paracrine secretion. Accumulating evidence supports the beneficial effects of EVs for tissue regeneration. In this study, discarded human heart tissues were used to isolate human heart-derived extracellular vesicles (hH-EVs). We used nanoparticle tracking analysis (NTA) and transmission electron microscopy (TEM) to physically characterize hH-EVs and mass spectrometry (MS) to profile the protein content in these particles. The MS analysis identified a total of 1248 proteins. Gene ontology (GO) enrichment analysis in hH-EVs revealed the proteins involved in processes, such as the regulation of cell death and response to wounding. The potential of hH-EVs to induce proliferation, adhesion, angiogenesis and wound healing was investigated in vitro. Our findings demonstrate that hH-EVs have the potential to induce proliferation and angiogenesis in endothelial cells, improve wound healing and reduce mesenchymal stem-cell adhesion. Last, we showed that hH-EVs were able to significantly promote mesenchymal stem-cell recellularization of decellularized porcine heart valve leaflets. Altogether our data confirmed that hH-EVs modulate cellular processes, shedding light on the potential of these particles for tissue regeneration and for scaffold recellularization.

## 1. Introduction

A wide variety of cell types, such as cardiomyocytes, endothelial cells, smooth muscle cells and cardiac resident stromal cells (CRSCs), such as fibroblasts and stem/progenitor cells, are found in the adult heart [1,2]. CRSCs have been isolated and characterized by the ex vivo culture of cardiac explants [3,4,5]. The conditioned media from cultured CRSCs exhibits cardioprotective and regenerative capabilities, being able to drive proliferation, cardiac differentiation and tube formation in endothelial cells [4]. Similarly, studies with cardiosphere-derived cells (CDCs) have also pointed out the beneficial effects on cardiac cells and heart function [6], CDCs lead to myocardial regeneration and functional improvement of the heart when injected into infarcted mice [7]. A key component of paracrine secretion is the heterogeneous population of small membranous particles, called extracellular vesicles (EVs), which are released by cells. According to their size and origin, EVs can be classified into exosomes, endocytic origin and diameter of 30-100 nm; microvesicles, plasma membrane budding origin and diameter of up to 1000 nm; or apoptotic bodies, cell fragments of irregular form and variable size of up to 5 µm [8]. EVs carry small bioactive molecules, such as lipids, DNA, RNA, miRNA and proteins [9]. Since they were first reported in 1964 [10], EVs have emerged as important components for intercellular communications under normal physiological and pathophysiological conditions [11,12].

Indeed, the potential use of EVs as therapeutic components has gained considerable interest, particularly for driving renal and cardiac repair [13,14,15]. However, there have been few studies on the regenerative properties of the vesicles derived from whole intact cardiac tissue obtained from explant of all cardiac regions. Recently, mesenchymal stem cell-derived extracellular vesicles were used to induce cell migration and vascularization into the acellular bone matrix [16]. Acellular matrices are scaffolds produced for tissue-engineering approaches such as the decellularization process. In this context, decellularized heart valves (DHV) are prostheses widely used to treat diseases that depend on valve replacement. However, cell repopulation into valvular scaffolds is limited and may result in implant degeneration [17]. Intensive efforts have been made to produce an ideal graft that has the capability to grow, repair, and remodel during the growth of the living recipient [18,19].

In the current study, human heart tissue discarded from a Brazilian valve bank was used to isolate human heart-derived extracellular vesicles (hH-EVs), which were characterized by nanoparticle tracking analysis (NTA) and proteomics. When assayed in vitro, hH-EVs have the potential to induce proliferation and angiogenesis in endothelial cells as well as stimulate “wound healing” and suppress the adhesion of human mesenchymal stromal/stem cells. Additionally, EVs could improve the recellularization of decellularized porcine heart valve leaflets with mesenchymal stem cells. Thus, hH-EVs have emerged as promising tools for scaffold functionalization and tissue regeneration.

## 2. Results

### 2.1. Explant Characterization: Cells Derived from Different Heart Regions Are Similar and Heterogeneous

The immunophenotypically characterization of migrating cell populations from human heart explant cultures (derived from at least two different donors) was performed to identify different cell lineages from distinct regions of the heart: left ventricular myocardium (LVM), left ventricular endocardium (LVE), right ventricular myocardium (RVM), right ventricular endocardium (RVE), right auricle myocardium (AUM), right auricle endocardium (AUE) and mitral valve leaflet (MTL). Surface markers associated with fibroblasts and mesenchymal stem cells (CD90, CD105, CD73), endothelial cells (CD31), smooth muscle cells or pericytes (CD146, CD140b), cardiac progenitor cells (CD117) and cardiac fibroblasts (DDR-2) were analyzed (Table 1). Cardiac-derived cells migrated from all heart regions studied were strongly positive for CD105 and CD73, with the percentage of positive cells being approximately 66% in LVM cells and over 90% in the other heart regions. On the other hand, CD90 was consistently expressed in less than 60% of the cell populations. CD140b-positive cells were more common in the myocardial population (avg = 73%) than in the endocardium or mitral leaflet (avg = 28%). All cellular content analyzed revealed a small percentage of putative progenitor (5.09%–24.95%) and endothelial (0.96%–3.16%) cells. DDR-2-positive cells and cardiac fibroblasts were also observed ranging from 4.98% (LVM) to 30.95% (LVE). The lack or low percentages of some cell surface markers do not necessarily indicate that these cells are absent but, more likely, were not able to migrate or proliferate in the explant culture.

### 2.2. Characterization and Prospective Functional Analysis of Human Heart-Derived Extracellular Vesicles (hH-EVs)

To evaluate the size and concentration of EVs extracted from cardiac explants, the samples from each donor were individually examined using NTA. NanoSight analysis showed that samples are formed by a heterogeneous population composed of particles of 30 nm to 400 nm (Figure 1A) with a very similar distribution (Figure 1B). Electron microscopy of EVs revealed the presence of spherical particles with double membranes and compatible sizes, as observed in NTA (Figure 1A).

To better characterize and shed light on the putative functions of hH-EVs, a mass spectrometry (MS) analysis of the gel fractioned-EV proteins of three heart donors was performed. A proteomic analysis was carried out with no distinction among heart regions, and EVs were obtained from heart explants mixed in equal amounts. To increase the confidence level of the proteomic analyses, we only considered proteins that had a false discovery rate (FDR) value ≤1% and were detected in at least two out of the three donors. The MS analysis identified a total of 1248 proteins, among which 523 were common to all three samples and 887 were detected in at least two samples (Figure 2A). Thus, the protein content was qualitatively similar among samples. The extracellular vesicle origin of the sample was indirectly tested by determining the percentage of proteins identified in our study that are included in the Vesiclepedia database (http://www.microvesicles.org/index.html#). Surprisingly, 93% of the 887 proteins identified here and used for further analyses, are part of the protein list in the Vesiclepedia dataset. The percentage of unique proteins in each sample was low, 5% for hHCM1, 7% for hHCM2 and 23% for hHCM3. Since our proteomic study is not saturating, the absence of a protein does not indicate that it is not present in the sample. To determine the putative function of the proteins identified in the hH-EVs, we conducted a gene ontology (GO) analysis with the proteins identified in at least two samples using g:Profiler (Appendix A) software and summarized the results using Revigo. Among the most significant biological process terms related to exocytosis, vesicle-mediated transport, regulation of cell death, cell-substrate adhesion and response to wounding were identified (Figure 2B). Concerning molecular function, proteins involved in binding, GTPase activity, transporter activity and oxidoreductase activity were identified (Figure 2C). Finally, the proteomic analysis detected tetraspanins (CD81, CD9, CD63), heat-shock proteins (HSP-90) and other proteins enriched in exosomes. As a result, the major GO cellular component was extracellular exosome, followed by cytoplasm, vesicle and other terms such as coated membrane vesicles (Figure 2D). Altogether, these data suggest that our samples are highly enriched in EVs. Additionally, taking into account that the hH-EVs were obtained from discarded human heart material and that these vesicles exhibit a protein signature that might have interesting properties regarding tissue engineering and/or regenerative medicine, these possibilities are worth further study.

### 2.3. hH-EVs Stimulate Directional Cell migration and Decrease the Adhesion Capabilities of Adipose-Derived Stem Cells (ADSCs)

To determine whether hH-EVs could influence the adhesion of HUVECs (human umbilical vein endothelial cells—Lonza Clonetics^®^, Rochester, NY, USA) and ADSCs (human adipose-derived stem cells—Lonza Clonetics^®^, Rochester, NY, USA), we evaluated the number of cells adhered using high-content imaging system, and the cell adhesion period was established after standardization (Appendix A). The results obtained with ADSCs showed an overall adhesion reduction induced by hH-EVs, but this reduction was significant only for LVE (Figure 3A). On the other hand, we did not observe any statistically significant difference in the number of attached HUVECs between the control groups (medium FBS-deprived) and the hH-EV groups (Figure 3B). Similarly, we also evaluated the migration capacity of ADSCs and HUVECs influenced by hH-EVs. When the assay was performed with ADSCs, the number of cells in the wound area after 24 h was significantly augmented with the addition of AUE-derived vesicles to the medium (Figure 4A,C). This assay did not reveal any significant differences in HUVEC migration capabilities induced by cardiac EVs (Figure 4B,C).

### 2.4. hH-EVs Stimulate Proliferation and the in Vitro Angiogenesis of Human Umbilical Vein Endothelial Cells (HUVECs)

To evaluate the proliferation-promoting activity of hH-EVs, an assay was performed using EdU, a thymidine analog that was incorporated into the cells during 24 h under EV stimulation. The results obtained showed that hH-EVs were not able to induce mesenchymal stem cell proliferation (Figure 5A,C). On the other hand, all samples of EVs significantly induced the cell proliferation of HUVECs in vitro, except for the LVE sample (Figure 5B,C). Considering the endothelial cell proliferation induced by hH-EVs, we performed an in vitro assay to verify the angiogenic potential of cardiac EVs on HUVECs. Our results showed that hH-EVs derived from all heart regions were able to significantly induce tube-like structures after 6 h of culture on the Matrigel layer compared with the control medium without hH-EVs (Figure 6A). Surprisingly, the in vitro angiogenic effects reached levels and quality consistent with the gold standard control (5% fetal bovine serum (FBS)). During the time course of the experiment, tube-like structures decreased. However, after 12 h, the number of meshes induced by LVE, AUE, RVE, RVM and MTL extracellular vesicles was significantly higher than the control (Figure 6B). Although, after 24 h, the number of capillary-like networks stimulated by hH-EVs remained higher than that stimulated by the control, and the differences were not statistically significant (Figure 6C).

### 2.5. Effect of Left Ventricular Endocardium Extracellular Vesicles (LVE-EVs) on Leaflet Scaffold Recellularization

Before the valve scaffold recellularization experiments, we confirmed whether the leaflets were satisfactorily decellularized through the optical evaluation of nuclei presence/absence by using bright field and fluorescence microscopy (Appendix A). No nuclei were observed in any of the leaflet scaffolds used in our study. When ADSCs were cultured under standard conditions, after 24 h of cell-scaffold interactions, a layer of cells was found attached to the scaffold surface. However, when scaffolds were previously functionalized with LVE-EVs, a significant reduction in the number of cells adhered to the scaffold surface was observed (Figure 7A; Appendix A). Considering the observed effects of hH-EVs on ADSC migration on plastic plates (Figure 4), we wondered whether hH-EVs could potentiate ADSCs to colonize the decellularized scaffolds once these cells had become adhered. To this end, unfunctionalized scaffolds were transferred to a low-binding plate and cultured with 10 µg/mL of LVE-EVs. Interestingly, after 3 and 7 days of culture, the ADSCs under EV stimulation were able to colonize the leaflet scaffolds more efficiently than the ADSCs under control conditions (Figure 7B; Appendix A).

## 3. Discussion

In the present study we used human cardiac tissues explanted from cadaveric donors to isolate EVs. This material, which would be discarded, can be an interesting tool for understanding paracrine signaling in the heart and examining a putative agent in cell-free therapy approaches. Whereas the majority of studies involving cardiac-derived EVs were performed with isolated cells, we focused on the study of EVs derived from heart fragments that contain a complex network of cells.

The human heart comprises of four distinct chambers (two atria and two ventricles) arranged in three layers: pericardium, myocardium and endocardium [20]. A number of animal and human studies have been conducted to estimate the abundance of cardiac resident cells [2,21,22]. Currently, the most abundant cell types in mammalian hearts are cardiomyocytes (25%–35%) [21], fibroblasts (< 20%) and endothelial cells, which were recently identified as the most frequent cell type in mice hearts (60%) [2]. In previous studies, we demonstrated that CRSCs isolated by explant tissue culture consist of a heterogeneous population with the characteristics and differentiation potential of mesenchymal stem cells (MSCs) [4]. Here, we separated the heart layers and chambers to obtain explant characterizations for each cardiac region. Phenotypic profiling indicated that the cells that migrated from explants are a heterogeneous population that was mostly similar between the cardiac regions studied. The majority of cells were strongly positive for fibroblasts and mesenchymal stem cells and few cells were positive for endothelial and progenitor markers. This finding is in accordance with studies that isolate cardiac cells for cardiospheres culture and expansion [23]. Additionally, CD140b-positive cells seemed to be more abundant in the outgrown population of myocardium than endocardium samples. It is important to stress that the immunophenotypic characterization was performed on the migrating cell population from the human heart explants and not necessarily reflects the cell content of these explants.

So far, many efforts have been directed towards the study of cardiac-derived extracellular vesicles. Previous studies have been characterized the particles released by cardiomyocytes [24,25], cardiac-derived progenitor cells [26,27] and also cardiac cells derived from induced pluripotent stem cells (iPS) [28]. In the present study, for the first time, the EVs released from explanted human cardiac tissue were isolated. The EV population evaluated was derived from the pool of cell types present in each cardiac explanted tissue region, not from isolated cardiac cells. Nanoparticle tracking analysis and transmission electron microscopy collectively revealed that the purified hH-EVs are a heterogeneous population with a wide size range suggesting the presence of both exosomal and non-exosomal EVs. Furthermore, the hH-EVs showed similar sizes and distributions between cardiac samples. The mechanical disruption to obtain the explant and the culture might produce cellular debris and apoptotic bodies; however, most of them would be filtered out during our experiment. Thus, we believed that our hH-EVs preparations contain little to no contamination.

To further study the putative functions of the hH-EVs, a proteomic analysis was performed. We identified a total of 1248 proteins of which 887 were shared between at least two samples. Not surprisingly, the functional enrichment analysis revealed that most of the proteins were related to exosomes and vesicles. Consistently, major biological process GO terms were associated with exocytosis and vesicle-mediated transport. In addition, terms such as regulation of cell death, cell-substrate adhesion and response to wounding were found and may help to explain some effects observed in functional experiments. In fact, in assays conducted with stem cells, the AUE-derived EVs promote significant augmentation in ADSC “wound healing”, also, the cell migration appeared to be facilitated, although not significantly, by LVE and LVM EV samples. These results are consistent with previous findings in which cardiac progenitor cell (CPC)-derived EVs reduce scarring after the induction of myocardial infarction in rats [26]. Furthermore, some proteins involved in cell migration, such as CD47 [29], ITGA1 [30] and ILK [31] were identified in hH-EVs through mass spectrometry analysis (Appendix A). On the other hand, an overall adhesion reduction was observed in cultures of ADSCs supplemented with hH-EVs; however, the significant impairment in ADSC adhesion was only observed with EVs from LVE samples.

The effect of hH-EVs in reducing ADSC adhesion was an unexpected finding since the proteomic analysis revealed several cell-adhesion related proteins in hH-EVs as integrins and cadherins, among others (ITGA1, ITGA5, ITGB1, ILK, VTN, CDH13 and CTNNB1). However, it was recently demonstrated that endothelial-derived EVs can attenuate monocyte adhesion to HUVECs by negative regulation of adhesion proteins expression [32]. As previously shown [33], this effect was related to the microRNA (miR)-222 transference. Thus, further analysis like hH-EVs RNA sequencing would be necessary to elucidate this apparently controversial result.

In contrast, the hH-EVs did not affect the adhesion and wound-healing capabilities of endothelial cells. However, after 24 h of treatment, hH-EVs induced significant HUVEC proliferation compared to the control. Additionally, a time-course experiment was performed to determine the hH-EV potential to induce tube-like structures. The angiogenic potential was notorious in all EV samples, and the quality of tube structures was observed to be similar to the gold standard condition. Interestingly, hH-EVs isolated from the endocardium appeared to be more efficient in sustaining tube-like structures. Previous studies have demonstrated that exosomes released by H9C2 cardiomyoblasts were able to induce proliferation and stimulate angiogenesis in HUVECs [34]. Indeed, angiogenesis appears to be modulated by EVs derived from different cell types, depending on the EV content and surface molecule expression [35]. Although we did not find any GO terms associated with angiogenesis, the protein β1 integrin was identified in hH-EVs (Appendix A). This molecule was previously reported as a mediator of the proangiogenic effect caused by HUVEC-derived EVs [36]. In addition, other mechanisms are involved in angiogenesis induced by EVs, such as microRNAs and lipid transference [35] which could not be evaluated in this study. It is also worth mentioning that studying the effects of hH-EVs on ADSC and HUVEC not necessarily represent the precise physiological targets of hH-EVs.

These findings prompted us to verify a putative application for hH-EVs. Tissue engineering is constantly innovating to produce scaffolds for many applications [37,38,39]. A technique widely under investigation is cell-free scaffold production. In this study, we used a previously standardized methodology [40,41] to generate decellularized porcine heart valves. Subsequently, the leaflets were removed and repopulated with ADSCs. Although DHVs show promising results for valve replacement, the clinical application of these scaffolds remains a challenge because of the limited cell repopulation into the decellularized natural matrix [17]. One strategy to facilitate cell recruitment into the DHV is the use of bioactive molecules [42,43,44]. Complete recellularization of the acellular scaffold is highly desirable because it can modulate the host response to transplanted tissue [45] and inhibit tissue calcification [46]. In our experiments, LVE-EVs were used as bioactive molecules in ADSC repopulation. In fact, the LVE-EVs significantly reduced stem cells adhesion to the valve matrix, corroborating the results of functional assays. However, when LVE-EVs were used to stimulate ADSC repopulation after cell adhesion on the leaflet surface, a significant increase in recellularization was observed, probably by allowing a higher rate of cell interiorization into the leaflet scaffolds. Unfortunately, because of the limited availability of hH-EVs, we could not test all samples in scaffold repopulation or diversify the concentrations tested. Therefore, some issues remain to be addressed. For instance, the proangiogenic potential of hH-EVs can be interesting to promote the formation of vascular structures, which is critical for many tissue engineering strategies.

In conclusion, we have demonstrated that hH-EVs comprise both exosomal and non-exosomal particles that contain a set of proteins advantageous for tissue regeneration approaches. In this study we used a post-fabrication approach to improve the functionality of the DHV through the delivery of a biological agent (hH-EVs) obtained from a discarded material from the preparation for transplantation of human heart valves. hH-EVs were able to modulate proliferation, wound healing, adhesion and angiogenesis differently depending on the cell type. The beneficial effects of hH-EVs have shed light on the regenerative properties of these particles and their applicability for scaffold modification.

## 4. Materials and Methods

### 4.1. Ethics Statements

The cardiac tissues used in this study were harvested from 8 human cadaveric donors at the Human Tissue Bank (PUCPR, Paraná, Brazil). The tissues were found to be unsuitable for therapeutic use and were discarded after heart valve dissection. According to the current Brazilian legislation (Ordinance nº 2048, Section XI of transplants, Clause 478), this material was approved for scientific research purposes. This study was conducted in accordance with the Declaration of Helsinki. For experiments using porcine heart valves, the tissues were obtained from a local slaughterhouse (FrigoKeller LTDA—Paraná, Brazil). All procedures were performed after approval by the Human and Animal Ethics Committees of Pontifical Catholic University of Paraná (approval number 1.455.773).

### 4.2. The Culture and Characterization of the Explants

For tissue explant cultures, human cardiac fragments of the right auricle, right ventricle, left ventricle and mitral valve leaflet were obtained from cadaveric donors. Initially, the endocardial tissue was released from the myocardium of auricle and ventricle fragments. Next, the specimens were processed separately in samples from the left ventricular myocardium and endocardium (LVM, LVE), the right ventricular myocardium and endocardium (RVM, RVE), the right auricle myocardium and endocardium (AUM, AUE) and the mitral valve leaflet (MTL).

After separation, the tissues were manually minced into 1–2 mm^3^ fragments as previously described [3]. Briefly, the explants were adhered to flasks coated with collagen film (Sigma, Ronkonkoma, NY, USA) and cultured in rich medium to allow growth of a wide variety of cell types. MegaCell™ (Sigma, USA) medium was supplemented with 5% fetal bovine serum (Gibco™ Invitrogen Corporation, Big Cabin, OK, USA), 2 mM l-glutamine (Gibco^TM^ Invitrogen Corporation, USA), 5 ng/mL basic fibroblast growth factor, 0.1 mM β-mercaptoethanol, 1% nonessential amino acids, 100 IU/mL penicillin and 0.1 mg/mL streptomycin (Sigma, USA), at 37 °C and 5% CO_2_. Then, cells that migrated from the fragments were dissociated using 0.25% trypsin-EDTA and replated at a density of 0.2–0.5 × 10^4^ cells/cm^2^.

To characterize the explants obtained from different heart regions, migrated and cultured cells at passages 2–3 were immunophenotypically characterized by flow cytometry with fluorochrome-conjugated antibodies against the following human proteins: CD90 (1:20), CD105, CD73, CD31 (1:50), CD117 (1:20) (eBioscience, San Diego, CA, USA), CD140b, CD146 (1:50) (BD Biosciences, San Jose, CA, USA), and DDR-2 (1:50, Abcam, Cambridge, UK). Mouse isotype IgG1 antibodies for each fluorochrome were used as controls (BD Biosciences). The data were acquired with a FACSCanto II flow cytometer (BD Biosciences) and analyzed by FlowJo software version 10.0.8r1 (http://www.flowjo.com/).

### 4.3. Isolation of Heart-Derived Extracellular Vesicles

To isolate human heart-derived extracellular vesicles (hH-EVs) the tissues were processed as described for the explant characterization (Section 4.2) (Figure 8); however, after dissociation, the heart fragments were cultured at 37 °C and 5% CO_2_ in 5 mL of Dulbecco’s Modified Eagle’s Medium (DMEM) supplemented with 100 U/mL penicillin, 0.1 mg/mL streptomycin and 10% fetal bovine serum depleted of vesicles (dFBS). The dFBS was obtained from the supernatant of FBS ultracentrifuged under the same conditions used for the isolation of vesicles to avoid cross-contamination (Appendix A).

After 24 h of cardiac tissue incubation, the culture supernatants were collected, filtered on a 40 μm porosity cell strainer and centrifuged two times at 700 g for 5 min and 4000× *g* for 20 min followed by 0.22 μm membrane filtration. The supernatants were ultracentrifuged for 1 h and 20 min at 100,000× *g* and the pellets were resuspended in phosphate-buffered saline (PBS) and stored at 4 °C. This procedure of supernatant collection/storage at 4 °C was followed by culture medium (supplemented with dFBS) replacement to ensure three consecutive isolations of hH-EVs. After the third day of EV collection, all obtained samples were submitted to ultracentrifugation at 100,000× *g* for 2 h. Finally, the purified hH-EVs were resuspended in small volumes of PBS and the protein concentration of the samples was determined using a Qubit^®^ fluorometer (Life Technologies™, Invitrogen, NY, USA). The hH-EV samples were stored at −80 °C until further used.

Equal amounts of the isolated hH-EV samples derived from at least 3 donors were mixed into pools according to the seven heart tissue regions evaluated (LVM, LVE, RVM, RVE, AUM, AUE, MTL; Table 2). The pools were concentrated by centrifugal filters (Amicon^®^ Ultra—0.5 mL 30 K, Millipore, Tullagreen, Ireland) at 0.5 μg/μL and used to evaluate their effects on cell adhesion, proliferation, migration and angiogenesis.

### 4.4. hH-EV Characterization by Nano-Tracking Analysis, Transmission Electron Microscopy (TEM) and Mass Spectrometry (MS) Analysis

The physical characterization of hH-EVs was conducted using NTA and transmission electron microscopy (TEM). To record data on hH-EV concentration and size, samples derived from each cardiac region were diluted 20-fold in PBS and analyzed using NanoSight LM10 (NanoSight Ltd., Amesbury, UK) and NTA 3.6 analytic software. The TEM analysis was performed by loading 15 µg of hH-EVs onto 300-mesh nickel/formvar-coated grids (Electron Microscopy Science, Washington, PA, USA) for 1 h. After that, the grids were fixed with 4% paraformaldehyde (Electron Microscopy Science) and postfixed in 2.5% glutaraldehyde (Sigma-Aldrich, St. Louis, MO, USA) plus 0.1 M cacodylate for 10 min each. The grids with hH-EVs were washed with ultra-pure water and stained for contrast using 2% uranyl acetate for 10 min. The grids containing samples were analyzed on a transmission JEOL JEM-1200 EX II electron microscope (JEOL Ltd., Tokyo, Japan) (Electron Microscopy facility, Instituto Carlos Chagas) operating at an acceleration voltage of 80 kV.

To disclose the general protein composition present in the hH-EVs, explants from myocardial auricle and ventricle regions were cultured at a ratio of 1:1 under the same conditions as described above. Samples from three different donors were obtained. The hH-EV isolation was conducted as described in Section 4.3. Then, proteomic analysis was carried out with 30 µg of hH-EVs per donor as previously described by our group [47].

### 4.5. ADSC and HUVEC Cultures

For the assessment of hH-EVs function in cell signaling in vitro, two different cell lineages were used: ADSCs maintained in DMEM supplemented with 100 U/mL penicillin, 0.1 mg/mL streptomycin and 10% FBS; and HUVECs cultured in endothelial cell growth medium (EBM-2) supplemented with hydrocortisone, hFGF, VEGF, IGF, ascorbic acid, hEGF, 30 mg/mL gentamicin and 15 μg/mL amphotericin. All cell cultures for the different in vitro assays were maintained at 37°C and 5% CO_2_.

### 4.6. Cell Proliferation Assay

To evaluate the potential of hH-EVs to induce ADSC and HUVEC proliferation, the cells were seeded at 5 × 10^3^ cells/well on to 96-well plates in 50 µl of DMEM or EBM-2 deprived of FBS. After 24 h, the medium was removed, and 10 µg/mL hH-EVs and 5 μM EdU diluted in 50 μL of medium deprived of FBS were added to the wells for 24 h. EdU (5-ethynyl-2′-deoxyuridine) incorporation into DNA synthesized over 24 h was detected with the Click-iT^®^ EdU Alexa Fluor^®^ 594 Imaging Kit (Thermo Fischer, Holtsville, NY, USA), according to the manufacturer’s instructions. In order to quantify the cell proliferation, 3 pictures of each well (10× magnification) were recorded using DMI6000B microscope (Leyca Microsystems, Wetzlar, Germany), and the percentage of cells positively labeled for EdU was analyzed by NIH ImageJ software v. 1.45s (National Institute of Health, Bethesda, MD, USA). Before the cell proliferation, adhesion, scratch-wound and in vitro angiogenesis experiments, the cells were synchronized by 24 h serum starvation. Control conditions were evaluated using medium deprived of FBS.

### 4.7. Cell Adhesion Assay

To evaluate whether hH-EVs could affect cellular adhesion, suspensions containing 5 × 10^3^ cells plus 10 μg/mL hH-EVs were plated onto 96-well plates and incubated for 20 min at 37 °C and 5% CO_2_. Immediately thereafter, the plates were subjected to 100 rpm agitation to promote nonadherent cell removal. The remaining cells were fixed with 4% paraformaldehyde and stained with 0.05 μL/mL DAPI (4′,6- diamidino-2-phenylindole). The quantification of the adhered cells was performed by an Operetta HTS imaging system (PerkinElmer, Waltham, MA, USA) at 10× magnification with 9 fields of view.

### 4.8. Scratch Wound Assay

The effect of hH-EVs on ADSC and HUVEC migration was measured by scratch wound assay. The cells were cultured in 96-well plates until confluence. Subsequently, a single scratch was created on the monolayer using a p200 micropipette tip. Next, the cells were washed two times with BSS-CMF and then 50 µl of medium containing 10 μg/mL hH-EVs was dispensed above the cultures. After 24 h, the cells were stained with crystal violet. For HUVECs, the percentage of wound closure was calculated by comparing the scratched area at 24 h with the initial scratched area (0 h). On the other hand, because the ADSCs spread throughout the scratch area, it was difficult to quantify the area that remained opened after migration. Thus, wound closure was calculated by comparing the number of cells at and out of the scratched area (100% closure) at 0 h and after 24 h. Quantitative analysis was performed with pictures (4× magnification) acquired using GmbH 37081 microscope (Carl Zeiss MicroImaging, Gottingen, Germany) and analyzed with NIH ImageJ software v. 1.45 s (National Institute of Health, USA).

### 4.9. In Vitro Angiogenesis Assay

To evaluate the potential of hH-EVs to generate tubule-like structures in vitro, 2 × 10^4^ HUVECs were plated in the wells of Matrigel^®^-coated 96-well plates (Matrix Basement Membrane, New York, NY, USA). The cells were cultured in 50 μL of EBM medium deprived of FBS and supplemented with 10 μg/mL of hH-EVs. The progress of capillary-like formation was recorded using GmbH 37081 microscope (Carl Zeiss MicroImaging, Gottingen, Germany). Each well was photographed (4× magnification) to register the entire network at 6, 12 and 24 h after plating. The number of meshes was quantified manually.

### 4.10. Decellularization of Porcine Heart Valves

Three porcine hearts (*Sus domesticus*) were obtained from a slaughterhouse, decontaminated in antibiotic solution (RPMI, streptomycin 1 mg/mL, penicillin 1 mg/mL, amphotericin 1 mg/mL) at 4 °C for 24 h and used for pulmonary valve dissection. The isolated valves were decellularized using 0.1% SDS and 0.02% EDTA solution for 24 h, as described previously [41]. To confirm the decellularization process, the valve leaflets were removed, fixed in 4% paraformaldehyde, embedded in paraffin and stained with 0.05 μL/mL DAPI or H&E (hematoxylin–eosin).

### 4.11. In Vitro Porcine Valve Leaflet Scaffold Recellularization

Decellularized leaflets were cut into 3 mm² fragments. To investigate the effects of hH-EVs on cell adhesion to the scaffold, EVs isolated from the left ventricular endocardium (LVE-EVs) were used. First, the fragments were coated with 30 µl of DMEM containing 10 µg/mL of LVE-EVs. Then, the scaffolds were maintained in hanging drop plates (Nunclon™ Surface, NUNC, Denmark) for 2 h at 37 °C. Then, the plates were inverted and incubated for additional 2 h under the same conditions. This step was established to allow the interaction between LVE-EVs and the scaffolds. Next, the functionalized leaflet scaffolds were incubated with 30 µl of 5 × 10^4^ ADSC at 37 °C and 5% CO_2_. The 3D cultures were maintained in hanging drop plates for 24 h and then processed to measure cell adhesion as described below.

To predict the effects of LVE-EVs after ADSC adhesion, we conducted an experiment with unfunctionalized leaflet scaffolds. The fragments were prepared as described previously (30 µl of DMEM, 37 °C, 4 h) but without LVE-EVs. Subsequently, the leaflet scaffolds were cultured with 5 × 10^4^ ADSCs in 30 µl of DMEM at 37 °C and 5% CO_2_ and maintained in hanging drop plates for 24 h. Next, the fragments were transferred to ultralow attachment 6-well plates, and 10 µg/mL of LVE-EVs were added to each well. After 3 and 7 days of culture, the samples were collected, fixed in 4% PFA and embedded in Tissue Tek O.C.T. compound.

The samples were sectioned into 15 µm slices and stained with DAPI. The histological preparations were photographed (10× magnification) to register the entire fragment. To determine the percentage of recellularization, the number of cells in each slice was quantified manually, and the area of fragment was defined using NIH ImageJ software v. 1.45 s.

### 4.12. Statistical Analysis

The experimental procedures were analyzed for statistical significance with GraphPad PRISM 6.0 software (GraphPad Software, San Diego, CA, USA), using Student’s *t*-test or one-way analysis of variance (ANOVA), followed by Dunnett’s multiple comparisons. The differences between means were considered significant if *p* ≤ 0.05. The results are expressed as the means ± standard error of the mean (SEM).

## Figures and Tables

**Figure 1 ijms-20-01279-f001:**
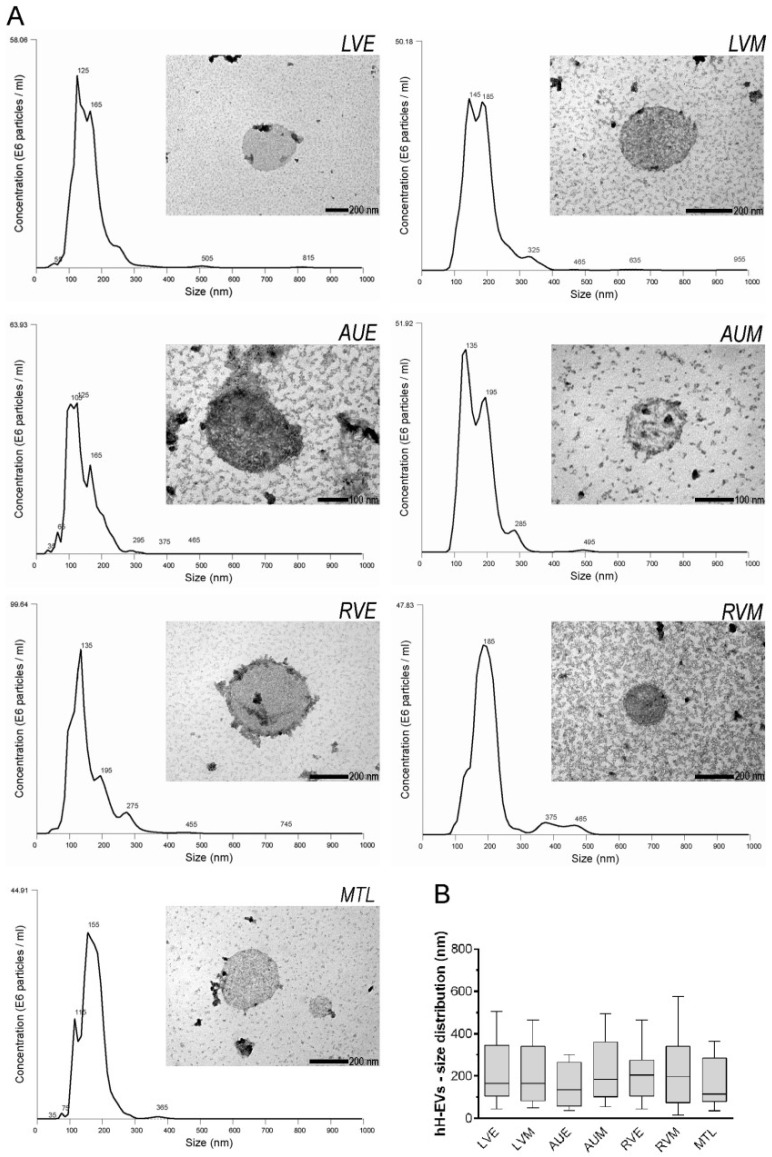
Characterization of hH-EVs. (**A**) Representative graphics from nanoparticle tracking analysis (NTA) and transmission electron microscopy (TEM) images of extracellular vesicles derived from cardiac regions: left ventricular endocardium (LVE), left ventricular myocardium (LVM), right auricle endocardium (AUE), right auricle myocardium (AUM), right ventricular endocardium (RVE), right ventricular myocardium (RVM) and mitral valve leaflet (MTL). (**B**) Analysis of the size and distribution of samples individually examined using NTA.

**Figure 2 ijms-20-01279-f002:**
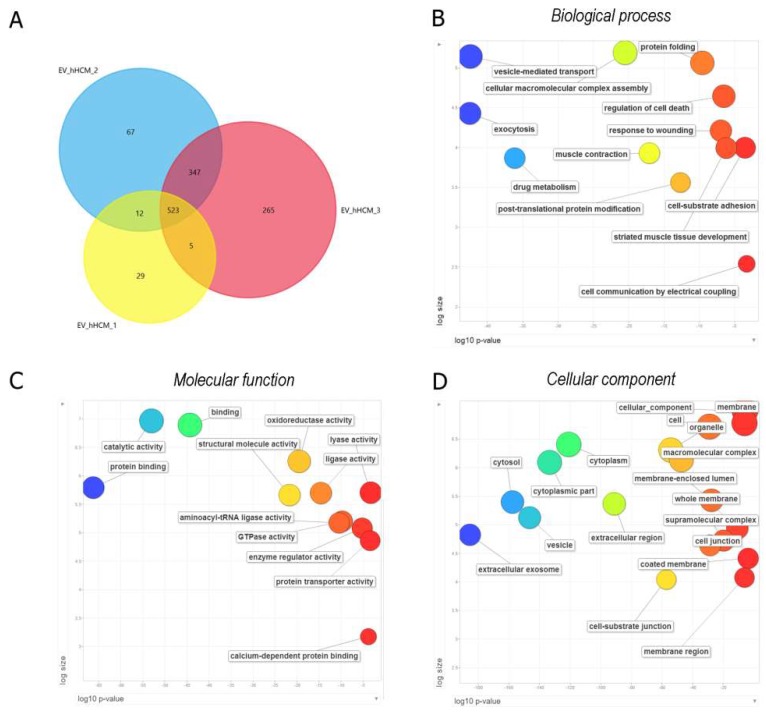
Protein identification and functional enrichment analysis of human heart-derived extracellular vesicles (hH-EVs). (**A**) Venn diagram of the proteins identified in three different samples. The diagram shows an overlap of the proteins that were common between the samples analyzed. (**B**–**D**) Gene ontology enrichment analysis summarized and visualized as a scatter plot using Revigo. The GO terms were ordered in relation to the p-value (*x*-axis) obtained from the GO term enrichment analysis and the frequency of GO terms in the Gene Ontology Annotation Database (*y*-axis). The Venn diagram was generated using Funrich software, and the GO analysis was conducted with g:Profiler software.

**Figure 3 ijms-20-01279-f003:**
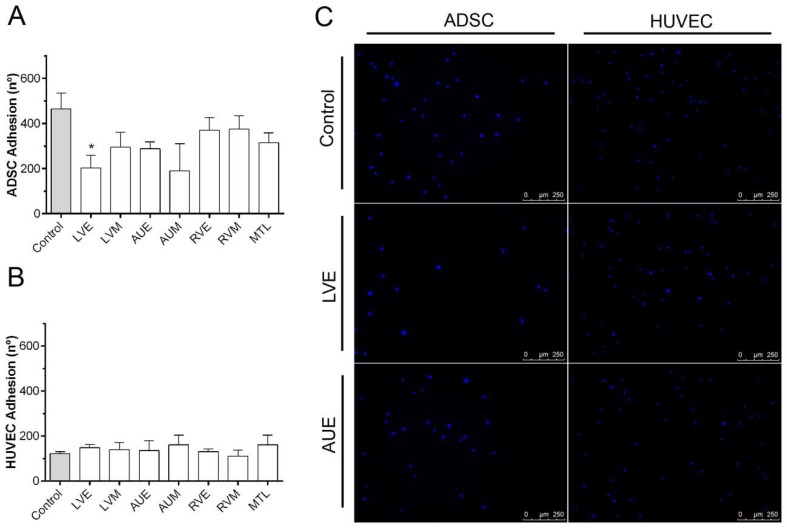
Influence of hH-EVs derived from cardiac regions on human adipose-derived stem cell (ADSC) and human umbilical vein endothelial cell (HUVEC) adhesion. Quantification of adhered (**A**) ADSCs and (**B**) HUVECs stimulated by hH-EVs. (**C**) Representative images of adhered cells after 20 min of treatment with extracellular vesicles derived from the left ventricular endocardium (LVE) and control without vesicles. Scale bar = 500 µm. * *p* < 0.05.

**Figure 4 ijms-20-01279-f004:**
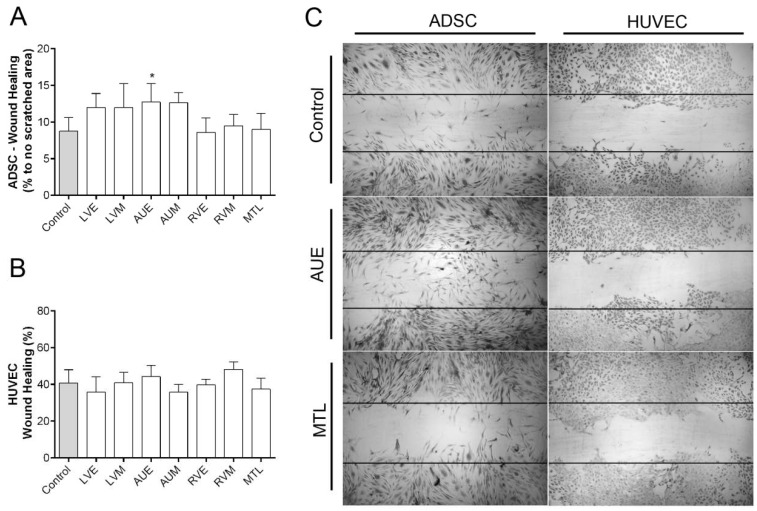
Influence of hH-EVs derived from cardiac regions on ADSC and HUVEC wound healing. (**A**) Quantitative analysis of the percentage of ADSCs in the scratched area after 24 h. (**B**) Percentage of wound closure by HUVECs after 24 h. (**C**) Representative images of wound healing stimulated by extracellular vesicles derived from the left ventricular endocardium (LVE) and the right auricle endocardium (AUE). Horizontal lines represent the initial scratched area (0 h), 4× magnification. * *p* < 0.05.

**Figure 5 ijms-20-01279-f005:**
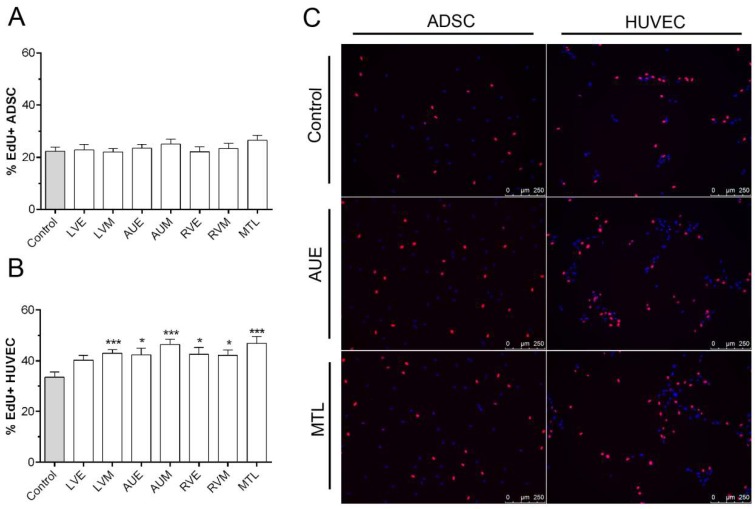
Influence of hH-EVs derived from cardiac regions on ADSC and HUVEC proliferation. Analysis of the percentage of EdU+ (**A**) ADSCs and (**B)** HUVECs cells after 24 h. (**C**) Representative images of EdU+ cells (red) stimulated by extracellular vesicles derived from right auricle endocardium (AUE) and mitral valve leaflet (MTL). * *p* < 0.05, *** *p* < 0.001.

**Figure 6 ijms-20-01279-f006:**
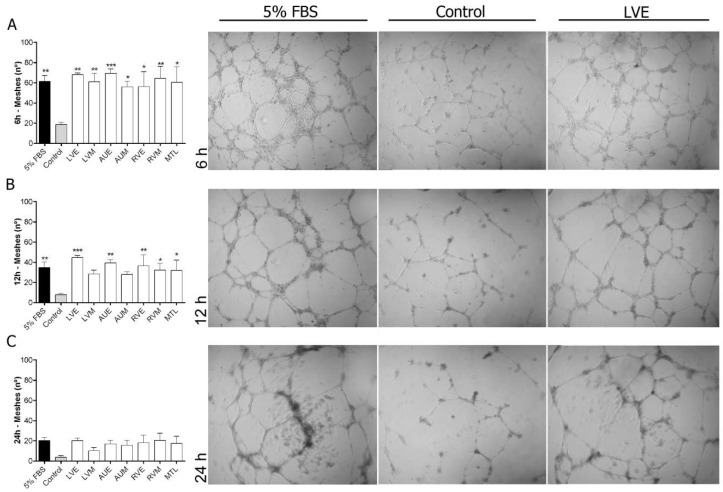
In vitro angiogenesis assay of HUVECs cultured for 24 h on a Matrigel layer under the influence of hH-EVs derived from cardiac regions. Representative images and analysis of the number of meshes formed after 6 h (**A**), 12 h (**B**) and 24 h (**C**). * *p* < 0.05 vs Control; ** *p* < 0.01 vs Control; *** *p* < 0.001 vs Control, 4× magnification.

**Figure 7 ijms-20-01279-f007:**
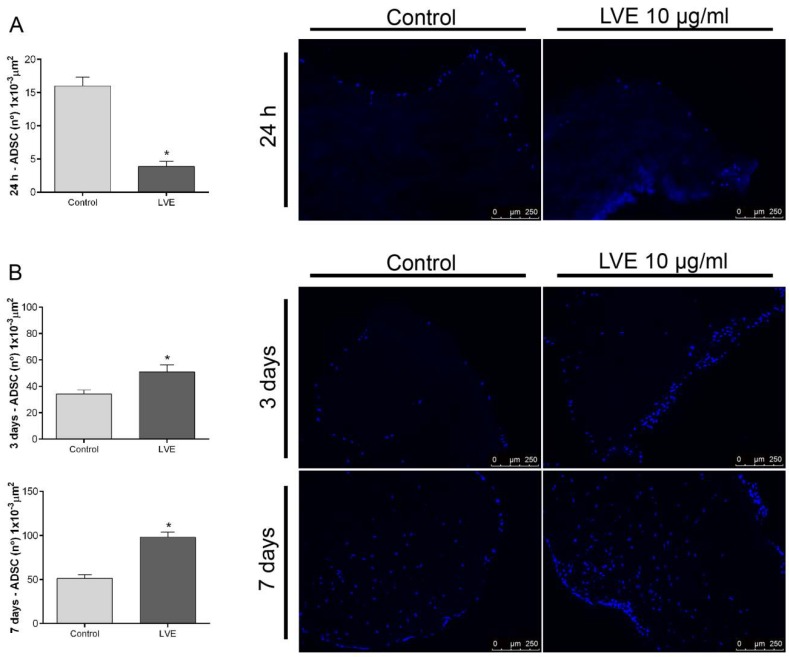
Extracellular vesicles derived from LVE affect scaffold recellularization. (**A**) Representative images and analysis of the number of cells/area of fragments previously coated with 10 µg/mL LVE-EVs and cultivated with ADSCs for 24 h. (**B**) Representative images and analysis of the number of cells/area of fragments cultivated with ADSCs for 24 h and then stimulated with 10 µg/mL LVE-EVs. The fragments were cultivated for 3 and 7 days. Unpaired *t* test * *p* < 0.05.

**Figure 8 ijms-20-01279-f008:**
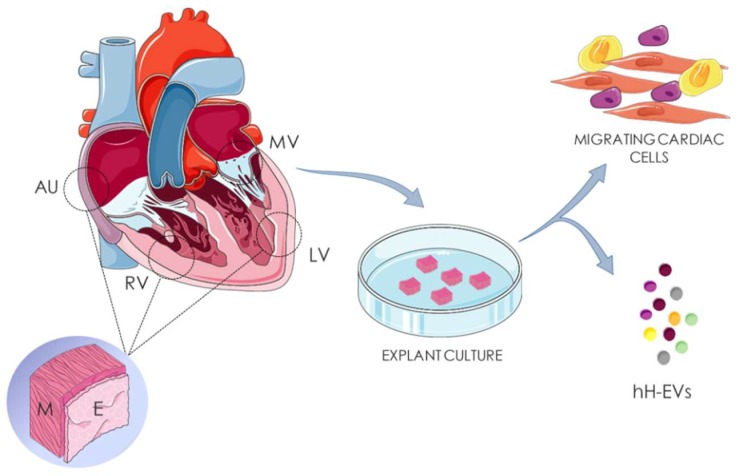
Isolation of heart-derived extracellular vesicles. Human cardiac fragments of right auricle (AU), right ventricle (RV), left ventricle (LV) and mitral valve (MV) were obtained from cadaveric donors. Endocardial (E) and myocardial (M) tissues of AU, RV, LV were separated before culture. The samples were dissociated in fragments of 3 mm² and culture for human heart-derived extracellular vesicles (hH-EVs) and cardiac cells isolation.

**Table 1 ijms-20-01279-t001:** Explant characterization.

Cell Markers	LVE	RVE	AUE	LVM	RVM	AUM	MTL
CD90	22 ± 8.3	17.15 ± 3.0	19.9 ± 6.0	27.1 ± 9.0	55.95 ± 4.5	31.2 ± 2.4	36.5 ± 20.4
CD105	92.05 ± 7.3	91.90 ± 7.4	84.65 ± 13.2	65.75 ± 17.0	94.75 ± 2.9	98.4 ± 0.2	98.65 ± 1.3
CD 73	97.7 ± 2.0	99.20 ± 0.3	91.15 ± 7.8	76.05 ± 23.4	95.4 ± 4.0	99.25 ± 0.5	97.6 ± 1.3
CD146	3.93 ± 3.5	1.25 ± 1.1	1.21 ± 0.6	0.91 ± 0.6	2.67 ± 0.8	1.35 ± 0.1	1.56 ± 0.2
CD140b	23.15 ± 10.5	32.80 ± 13.1	7.34 ± 1.0	53.60 ± 39.5	92.65 ± 1.1	67.05 ± 1.9	9.07 ± 1.9
CD117	6.20 ± 1.8	8.10 ± 2.3	11.4 ± 7.9	11.15 ± 1.1	24.95 ± 19.6	8.28 ± 3.6	5.09 ± 2.3
CD31	3.16 ± 1.4	2.13 ± 0.7	1.07 ± 0.4	1.71 ± 0.5	1.29 ± 0.3	0.96 ± 0.0	1.77 ± 0.1
DDR2	30.95 ± 24.4	22.65 ± 0.2	3.68 ± 2.3	4.98 ± 2.9	13.36 ± 13.4	17.15 ± 7.2	29.78 ± 22.5

**Table 2 ijms-20-01279-t002:** Clinical information about the heart donors and the hH-EV pool construction.

Subject	Gender	Age	Weight	Race	LVM	LVE	RVM	RVE	AUM	AUE	MTL
A	M	48	67	White	/	+	+	/	+	/	+
C	M	51	80	Black	+	+	+	+	+	+	/
D	F	50	60	Black	/	+	/	/	+	+	+
E	M	19	60	Mix of Races	+	/	/	+	/	+	/
F	M	42	115	White	/	/	/	+	/	+	+
G	F	56	56	Mix of Races	+	/	+	/	+	/	/

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
