# Peer review of "Human Heart Explant-Derived Extracellular Vesicles: Characterization and Effects on the In Vitro Recellularization of Decellularized Heart Valves"

_ijms, 2019, doi:10.3390/ijms20061279_

Reviewer 1 Report

Authors have addressed my comments

Author Response

Thank you very much for taking the time to review our manuscript.

Reviewer 2 Report

The authors have scholarly adressed the questions that had been raised. Maybe it would be more appropriate, in the title, to state that it is a "tissue engineering-targeted "approach than simply a "tissue engineering approach" because data on the latter are still limited in the paper.

Author Response

Reviewer comment:

The authors have scholarly adressed the questions that had been raised. Maybe it would be more appropriate, in the title, to state that it is a "tissue engineering-targeted "approach than simply a "tissue engineering approach" because data on the latter are still limited in the paper

Response to Reviewer #2:

As requested, we have included a more appropriate and specific title:

 ORIGINAL TITLE: Human Heart Explant-Derived Extracellular Vesicles in a Tissue Engineering Approach: A Prospective Study

NEW SPECIFIC TITLE: Human Heart Explant-Derived Extracellular Vesicles: Characterization and Effects on the in vitro Recellularization of Decellularized Heart Valves

Reviewer 3 Report

The authors have answered the critiques appropriately

Author Response

Thank you very much for taking the time to review our manuscript.

This manuscript is a resubmission of an earlier submission. The following is a list of the peer review reports and author responses from that submission.

Round  1

Reviewer 1 Report

In the present manuscript heart tissues was used to isolate human heart-derived extracellular vesicles (hH-EVs) from different anatomical regions. EV-derived from cells were functionally compared. The manuscript is easy to follow and experiments are well designed and described.

By looking at immunophenotypic characterizzation all cell linages appear similar in expression of surface markers but for RVM that showed higher expression of CD90, CD140b and above all CD117. However EVs from those cells do not differ among others in functional assays. Can Authors comment on that ? c-Kit and EVs from cKit cells have been shown to improve heart function and regeneration. Do authors have double staining showing whether cKit co-localize with other stem/mesenchymal marker?

Authors conclude that “The beneficial effects of hH-EVs have shed light on the cardioprotective properties….” However there is no evidence in this paper of cytoprotective role of hHEVs on cardiomyocytes.

How author explain the effects hEV on cell-adhesion. Is there any protein resulting from proteomic analysis that can be validated in this regard?

Reviewer 2 Report

In this study, the authors have used discarded human heart tissues to isolate human heart-derived extracellular vesicles (EVs). These vesicles have been characterized for their size, morphology and proteomic profile and then shown, in vitro, to reduce the adhesion of Human Adipose Derived Stem-Cells (ADSC), but not of Human Umbilical Vein Endothelial Cells (HUVEC). A similar pattern was observed for the migration capacity of the two tested cell types. Heart explant-derived EVs also increased the HUVEC proliferation and angiogenesis (assessed by the formation of tube-like structures after culture in Matrigel). Finally, porcine heart valves were decellularized, coated with EV and then recellularized with ADSC; it was then observed that EVs stimulated the recellularization of the scaffold surface.

While the use fo EV is a hot topic because of the possible therapeutic applications of these nanoparticles, the present study raises several serious methodological concerns.

1- It is not correct to state that "there have been little or no studies on the regenerative properties of whole cardiac tissue" in view of the numerious studies with cardiosphere-derived cells testing either the cells themselves or their secreted EVs. Cardiospheres are nothing else that chunks of right ventricular tissue containing a mix of different cell types, predominantly CD90-expressing stromal cells, just like in the present study. The authors can only be credited to have more specifically looked at tissues derived from the four cardiac chambers.  

2- In the phenotypic characterization of the heart explants, the authors have looked for markers of fibroblasts, mesenchymal stem cells, endothelial cells, smooth muscle cells, pericytes, cardiac  progenitor cells and cardiac fibroblasts. It is really surprising that they have not assessed the presence of markers for mature cardiomyocytes.

3- In the in vitro assay of cell migration, only EVs from the right atrium increased migration of ADSC while none of the tested EVs had an impact on the migration of HUVEC. The meaningfulness of this isolated finding remains uncertain and is never discussed. Furthermore, the interest of these assays is limited by the fact that EVs have been shown to have dose-dependent effects while this study tested a single dose (10 μg/ml). The rationale for the selection of this dose is not justified either. Furthermore, the angiogeneic capacity of EVs cannot be considered as a novel finding.

4- Heart fragments have been cultured in fetal bovine serum reported to be depleted of vesicles. However, Nanosight Tracking Analysis is fraught with many arfectacts and actually identifies particles, not necessarily vesicles. Thus, the protocol should have included control spectra (i.e., emitted by the purportedly vesicle-free serum) to eliminate the potentially confounding effects of contaminating vesicles originating from the culture medium. 

5- Althoug the title is appealing because of the mention of a "tissue engineering approach", this section of the paper is actually very limited. The heart valve scaffolds are reported to have been decellularized but Figure S2 shows a lot of DAPI stainings suggestive of the presence of nuclei. How do the authors reconcile this observation with their statement that "No nuclei were observed in any of the leaflet scaffolds used in our study"? Is there not an inversion in the labeling of the Figure?  Furthermore, in vitro, only EVs from the right atrium were able to increase migration of ADSC while in the recellularization experiments, increased recolonization was seen with EVs from the left ventricular endocardium. How can these seemingly discrepant data be reconciled? Finally, this increased recolonization by left ventricular endocardium-derived EVs is a merely observational finding. What is its physiological significance in these static conditions? What is its relevance to the function of a tissue-engineered heart valve?

 Reviewer 3 Report

Review of Leitolis et al.(submitted to Int’l J of Molecular Sci)

Strengths:

·         The identification of a novel effect of cardiac cell-derived extracellular vesicles (EVs) with potential therapeutic implications is exciting. One can envision the results of this line of study to be applicable in the clinic in the future.

·         The authors characterize cell populations and markers and EV morphology from each cardiac anatomical region as well as histological layer, thus addressing the biologic heterogeneity of these EV producers.

·         The effects of hH-EVs are studied both on mesenchymal stem cells and endothelial cells, and found to have different effects of both cell types.

Weaknesses:

·         While the proteomics and gene ontology studies of EVs the authors performed are interesting in identifying common pathways and biologic components present in EVs from different donors, no proteins are postulated as candidates in mediating the effects these EVs have on ADSCs or HUVECs.

·         In general, little mechanistic insight is provided in this paper to explain the observed effects induced by EVs.

·         Studying the effects of various hH-EVs on human cell lines (ADSCs and HUVECs) might not necessarily represent the physiological targets of hH-EVs accurately.

·         The EVs should be further characterized. As is, the authors only characterize their EV preparations by morphology and size distribution. While NTA and TEM are excellent tools to achieve this purpose, no EV markers (or markers of non-EV contaminants) are studied in these samples. For example, one wonders if they contain CD9, CD63, CD81, etc., tetraspannins which are markers of exosomes, associated with the producing cell population.

 Overall:

The paper by Leitolis et al shows that extracellular vesicles derived from particular cardiac anatomic regions have effects on mesenchymal stem cells and endothelial cells involved in wound healing and angiogenesis, which could have therapeutic implications. The paper could be strengthened by more thorough characterization of hH-EVs as well as future identification of candidate mediators of the observed effects.